# Understanding the Risks and Benefits of a Patient Portal Configured for HIV Care: Patient and Healthcare Professional Perspectives

**DOI:** 10.3390/jpm12020314

**Published:** 2022-02-19

**Authors:** Dominic Chu, David Lessard, Moustafa A. Laymouna, Kim Engler, Tibor Schuster, Yuanchao Ma, Nadine Kronfli, Jean-Pierre Routy, Tarek Hijal, Karine Lacombe, Nancy Sheehan, Hayette Rougier, Bertrand Lebouché

**Affiliations:** 1Department of Family Medicine, McGill University, Montreal, QC H3S 1Z1, Canada; dominic.chu2@mail.mcgill.ca (D.C.); moustafa.laymouna@mail.mcgill.ca (M.A.L.); tibor.schuster@mcgill.ca (T.S.); 2Canadian Institutes of Health Research Strategy for Patient-Oriented Research Mentorship Chair in Innovative Clinical Trials in HIV, Montreal, QC H4A 3J1, Canada; david.lessard2@mail.mcgill.ca (D.L.); kimcengler@gmail.com (K.E.); yuanchao.ma@muhc.mcgill.ca (Y.M.); 3Centre for Outcomes Research and Evaluation, Research Institute of the McGill University Health Centre, Montreal, QC H4A 3J1, Canada; nadine.kronfli@mcgill.ca; 4Department of Medicine, Division of Infectious Diseases and Chronic and Viral Illness Service, McGill University Health Centre, Montreal, QC H4A 3J1, Canada; jean-pierre.routy@mcgill.ca (J.-P.R.); nancy.sheehan@umontreal.ca (N.S.); 5Department of Mechanical Engineering, Polytechnique Montréal, Montreal, QC H3C 3A7, Canada; 6Division of Radiation Oncology, McGill University Health Centre, Montreal, QC H4A 3J1, Canada; tarek.hijal@mcgill.ca; 7Faculté de Médecine, Sorbonne Université, Inserm IPLESP, Hôpital St Antoine, APHP, 75012 Paris, France; karine.lacombe2@aphp.fr; 8Faculté de Pharmacie, Université de Montréal, Montréal, QC H3T 1J4, Canada; 9IMEA, Institut de Médecine et d’Épidémiologie Appliquée, F75018 Paris, France; hayette.rougier.sat@aphp.fr

**Keywords:** HIV, chronic disease, patient portal, patient engagement, qualitative research, self-management, health information technology (HIT), mobile health

## Abstract

Background: Like other chronic viral illnesses, HIV infection necessitates consistent self-management and adherence to care and treatment, which in turn relies on optimal collaboration between patients and healthcare professionals (HCPs), including physicians, nurses, pharmacists, and clinical care coordinators. By providing people living with HIV (PLHIV) with access to their personal health information, educational material, and a communication channel with HCPs, a tailored patient portal could support their engagement in care. Our team intends to implement a patient portal in HIV-specialized clinics in Canada and France. We sought to understand the perceived risks and benefits among PLHIV and HCPs of patient portal use in HIV clinical care. Methods: This qualitative study recruited PLHIV and HIV-specialized HCPs, through maximum variation sampling and purposeful sampling, respectively. Semi-structured focus group discussions (FGDs) were held separately with PLHIV and HCPs between August 2019 and January 2020. FGDs were recorded, transcribed, coded using NVivo 12 software, and analyzed using content analysis. Results: A total of twenty-eight PLHIV participated in four FGDs, and thirty-one HCPs participated in six FGDs. PLHIV included eighteen men, nine women, and one person identifying as other; while, HCPs included ten men, twenty women, and one person identifying as other. A multi-disciplinary team of HCPs were included, involving physicians, nurses, pharmacists, social workers, and clinical coordinators. Participants identified five potential risks: (1) breach of confidentiality, (2) stress or uncertainty, (3) contribution to the digital divide, (4) dehumanization of care, and (5) increase in HCPs’ workload. They also highlighted four main benefits of using a patient portal: (1) improvement in HIV self-management, (2) facilitation of patient visits, (3) responsiveness to patient preferences, and (4) fulfillment of current or evolving patient needs. Conclusion: PLHIV and HCPs identified both risks and benefits of using a patient portal in HIV care. By engaging stakeholders and understanding their perspectives, the configuration of a patient portal can be optimized for end-users and concerns may be mitigated during its implementation.

## 1. Introduction

Since the 2000′s, HIV clinical care has significantly improved as a result of effective and simpler combination antiretroviral therapies (cART) [1]. Once HIV infection is diagnosed and appropriately treated, people living with HIV (PLHIV) have a life expectancy comparable to that of the general population [2] and thanks to HIV undetectability, further transmission to others can be avoided [3]. However, therapeutic success requires ongoing treatment and self-management in a context where patients face greater risks for several comorbidities [4] and psycho-behavioral problems. Many of these difficulties rely on patient report for detection by healthcare professionals (HCPs) [5]. Patient engagement in care [6,7] is thus paramount to improve patient retention in care, cART adherence, and ultimately, health and quality of life [8,9,10]. Patient engagement refers to the degree to which patients are invested in their own healthcare [6] and partially depends on the quality of collaboration between PLHIV and HCPs [11]. Patient engagement can be influenced by various factors, such as the personalization of care, access to health education, health literacy, the level of patient empowerment, and patient-provider relationship dynamics [8,12]. Tools to foster the long-term self-management and engagement of PLHIV are thus needed.

With the growing digitalization of healthcare, patient portals may play a pivotal role in HIV self-management and the promotion of patient engagement in their HIV care [13,14,15,16]. Patient portals are connected platforms that allow patients access to their personal health information, including consultation notes, laboratory reports, education material, and their HCPs’ clinic contact information [17,18]. A patient portal may also facilitate the administration of patient-reported outcome measures (PROMs), which are self-reported patient questionnaires without provider interpretation or modification [19].

Patient portals have also been shown to support patients’ self-management and patient satisfaction as well as promote early identification of patients’ needs or problems [20,21]. Such benefits have been facilitated due to a patient portal’s ability to foster communication between PLHIV and their HCPs [22]. A prior study examining an electronic health record system in HIV care in San Francisco noted that most patients (>80%) found that accessing their personal health information supported their ability to manage their HIV diagnosis and other medical problems [23].

Patient portals have also shown high uptake in HIV care due to its perceived benefits and limited risks. Another study demonstrated portal use was high in PLHIV, especially those with increased health care needs [24]. The most utilized portal functions involved communicating with HCPs, receiving medication refills, and scheduling appointments. PLHIV also felt that by using a patient portal and accessing personal health information, they could better take medications as prescribed, feel more in control of their care, and better prepare for clinic visits [25]. Additionally, prior research suggests that confidentiality and privacy concerns of PLHIV using a patient portal in HIV care were limited, despite researchers’ concerns of HIV stigma [23,25]. However, such perceived risks and benefits have not been documented for PLHIV in Montreal, Canada and Paris, France as there is currently no available HIV-specific patient portal in use.

Our team’s intention is to successfully implement a stakeholder-approved patient portal into HIV care in Montreal, Canada and Paris, France. The diagnosis, treatment, and psychosocial circumstances of PLHIV are unique. For example, many PLHIV experience stigma, concerns for their privacy and the confidentiality of their diagnosis, and difficulties accessing primary care [26,27]. Thus, this study explores the perspectives of PLHIV and HCPs on the perceived benefits and risks of using a patient portal in HIV care.

## 2. Methods

### 2.1. Study Design

This study utilized a qualitative cross-sectional design based on semi-structured focus group discussions (FGD).

### 2.2. Context and Participants

Our team will implement an HIV-oriented configuration of a patient portal (Opal) into HIV care at the Chronic Viral Illness Service (CVIS) [28], which serves over 2000 PLHIV at the McGill University Health Centre (MUHC), one of the largest public HIV clinics in Montreal, Canada. Opal was originally co-designed by computer scientists, HCPs, and patients at the Cedars Cancer Centre of the McGill University Health Centre in Montreal, Canada [29]. Opal includes a smartphone application, which allows oncology patients to access their personal health information, including consultation notes, diagnostic test results, and treatment and planning information. It also provides patients with educational material tailored to their condition and stage of disease, a navigation tool to the hospital and to hospital clinics, an appointment calendar, and a check-in system. Opal also facilitates the administration of PROMs [19].

Our team’s research program includes the eventual implementation of a patient portal in other HIV services in Montreal as well as several hospitals in Paris, France, within the *Assistance Publique—Hôpitaux de Paris* (AP-HP) [30]. Hence, we recruited participants from the HIV/Infectious Disease Unit at Saint-Antoine Hospital (Paris, France), while emphasizing recruitment at the MUHC and other Montreal-based HIV-specialized clinics. Approval was granted from the Research Ethics Board of the MUHC (study number: 2020–5910). Following French legislation [31], no other specific ethics approval was required in France, and a confidentiality and data transfer agreement was formed between the MUHC and AP-HP.

This study was conducted with PLHIV and HIV-specialized HCPs. Inclusion criteria for PLHIV were to be: (1) 18 years old or older and (2) have no cognitive impairment. To be included, HCPs had to work at the CVIS, another HIV-specialized clinic in Montreal, or Saint-Antoine Hospital’s HIV/Infectious Disease Unit. Participants were excluded if they could not communicate with researcher team members in English or French. For PLHIV, a maximum variation sampling approach was used to capture a wide range of perspectives, including diversity within patient age, gender, and ethnicity [32]. PLHIV were identified by their HCPs during routine clinical appointments and were referred to the research coordinator who met with them individually to acquire informed consent. PLHIV were then invited to a focus group discussion by email or phone.

HCPs included physicians, nurses, social workers, pharmacists, and clinical care coordinators with at least six months of experience providing HIV care at the CVIS or at Saint-Antoine Hospital, as well as other Montreal-based clinics, including the Centre hospitalier de l’Université de Montréal, Clinique médicale du Quartier Latin, and Clinique médicale l’Actuel. HCPs were selected using purposeful sampling and were invited by email from the research coordinator.

### 2.3. Data Collection

FGDs are group discussions comprising between 3–8 individuals conducted by a neutral moderator and at least one observer [33]. We aimed to conduct up to ten FGDs, with a minimum of four per stakeholder group (PLHIV, HCPs) [33]. These targets were selected to ensure theme saturation. Empirical research has shown that three focus groups are sufficient to capture 80% of analytical themes [34].

Each FGD was conducted in-person in a conference room within each respective site by DL and BL. Prior to each FGD, the moderator gave a ten-minute PowerPoint presentation explaining each patient portal function considered (see Table 1). Refreshments were served at all FGD, and PLHIV from the MUHC were compensated $40, while HCPs and PLHIV from Saint-Antoine Hospital received no compensation, as per French legislation.

FGD were audio-recorded, were approximately two hours duration, and followed a semi-structured interview schedule focused on four main topics for both PLHIV and HCPs: (1) participant views on patient portal functions and recommendations for configuring them for HIV care; (2) perceived benefits of a patient portal for HIV care; (3) perceived inconveniences or risks of a patient portal for HIV care; and (4) other potential impacts of a patient portal for HIV care.

### 2.4. Data Analysis

FGD recordings were transcribed verbatim by a third party. The transcriptions were then inductively coded by two researchers, DC and DL, which involved creating novel nodes or headings within the text while reading the transcriptions [35]. All transcribed content was utilized and coded using NVivo 12 software [36], which facilitates the efficient management of vast amounts of qualitative data and groups individual codes into larger nodes. After initial coding, peer debriefing with both DC and DL involved reviewing the codes and creating higher order headings to condense observations within similar categories. Coding and collaboration between both researchers were performed to ensure credibility of the analysis [37]. Content analysis was then utilized to generate themes related to benefits and risks based on these nodes [38]. The coded data were then reviewed and examined by DL, MAL, and DC, who collaborated to generate, compare, and finalize themes. Resulting themes were then analyzed and described in detail, before further review with co-authors.

## 3. Results

A total of ten focus groups were conducted from August 2019 to January 2020: four with PLHIV and six with HCPs. For PLHIV, three were conducted at the MUHC (one in English, two in French), and one was led at Saint-Antoine Hospital (in French). For HCPs at the MUHC, one was held with physicians only (English) and two were held with other HCPs (French). Two FGDs were organized with HCPs from other Montreal clinics (French), and one at Saint-Antoine Hospital (French).

Table 2 presents the characteristics of both PLHIV and HCP participants. PLHIV (*n* = 28) consisted of 18 men, 9 women, and one individual who identified as other, whose ages ranged between 28 and 72, with a mean age of 48.8 years (SD = 11.8). For HCPs (*n* = 31), there were 10 men, 20 women, and one individual who identified as other. Their mean age was 46.6 years (SD = 11.4). These HCPs included 13 physicians, 8 pharmacists, 6 nurses, 2 clinical care coordinators, and 2 social workers.

The themes generated by content analysis and illustrative quotations are presented in Table 3. Five themes of possible risks and four themes of potential benefits of using a patient portal for HIV care were identified. The themes for potential risks and benefits were similar in both Canadian and French focus group discussions. A saturation of themes was achieved after eight FGDs, by which additional data did not lead to any new emergent themes [39]. Themes are described as follows.

### 3.1. Potential Risks of Using a Patient Portal

1. Breach of confidentiality: PLHIV and HCPs expressed concerns regarding the security of their health data and their control over it. Some members of both participant groups were worried that personal health information could be hacked, sent by mistake to the wrong people, or seen by others while they were using a patient portal on their phone. This was relevant to numerous functions including the calendar, notifications, diagnostic test results, account settings, consultation note sharing and access, text messaging, and PROMs. As a solution, PLHIV recommended new features including two-step authentication login methods, facial recognition, fingerprint or voice recognition, and email or text verification. PLHIV also suggested the option to deactivate some of the patient portals’ features in individual accounts or to identify in advance people with whom a patient could share their personal medical information. For this issue, HCPs suggested an automated log out function.

2. Stress or uncertainty: PLHIV were concerned that using a patient portal may be overwhelming due to complicated features as well as inaccurate, outdated, or difficult-to-interpret data. They were also concerned about HCPs’ ability to update a patient portal content regularly and PLHIVs’ proficiency in English or French medical information. These concerns pertained to the following functions: education material, texting, check-in, account settings, consultation notes, and diagnostic test results. As a solution, HCPs suggested customizing a patient portal through its account settings to fit each patient’s preferences and using a patient feedback survey to improve the portal’s usability. Additionally, both participant groups recommended HCPs provide explanations on laboratory results within a patient portal to simplify and clarify complex medical information. Lastly, HCPs also worried about inaccurate waiting times displayed on the check-in function and advised to only display waiting times if the function could account for delays.

3. Contribution to the digital divide: Participants mentioned that a patient portal could exacerbate inequitable access to healthcare as some PLHIV may not own the appropriate technology (e.g., a smartphone or internet access), thus limiting their access to a patient portal. As a solution, both participant groups recommended that a patient portal offer access through either a desktop computer or a smartphone.

4. Dehumanization of care: Both groups of participants were concerned that a patient portal could dehumanize care by reducing human interaction between PLHIV and HCPs. They felt opportunities to reassure, comfort, educate, and communicate in-person with PLHIV could be lost, resulting in heightened anxiety for PLHIV. Both PLHIV and HCPs highlighted the importance of HCPs’ role in meeting PLHIVs’ need to talk and share their experience. They felt that while educational videos or articles could support PLHIV, these resources should not replace in-person consultations, which strengthen the patient-clinician relationship. Moreover, PLHIV recommended creating a callback request tool or an emergency alert so PLHIV could request a face-to-face, phone or videoconferencing consultation when needed.

5. Increase in healthcare provider workload: HCPs and some PLHIV were concerned about expanding HCPs’ responsibilities to use a patient portal and manage additional patient needs and preoccupations. For example, several HCPs worried that PLHIV would attempt to reach them after hours through the HCP contact information provided with questions about consultation notes or diagnostic test results. Thus, some HCPs recommended a communication tool based on texting with nurses as first contacts, rather than calling.

### 3.2. Potential Benefits of a Patient Portal for HIV Care

1. Improvement in HIV self-management: Both PLHIV and HCPs perceived that a patient portal could improve self-management by increasing PLHIVs’ health literacy, cART adherence, self-management at home, and retention in care, especially with PLHIV with low adherence to cART. Functions associated with this perceived benefit included consultation notes, diagnostic test results, education material, and notifications. However, PLHIV and HCPs recommended that the calendar function be improved to allow users to re-schedule appointments and set reminders to take cART. They also wished for the addition of HIV-specific education material on monitoring HIV, HIV treatment, drug use, immigration, contact information, and community-based resources.

2. Facilitation of patient visits: Both participant groups mentioned that a patient portal could efficiently facilitate each step of a patient’s visit, including the commute from home to the clinic, patient check-in, the consultation process, and transfers to other specialties (e.g., endocrinology, hematology, etc.) in the health center. This benefit was supported by a navigational tool, virtual check-in, PROMs, and notifications. For participants from both end-user groups, these functions had the potential to support PLHIV to find the right clinic in a timely manner when using a navigational tool, identify patient priorities using PROMs, and efficiently send health information by sharing consultation notes to non-HIV HCPs (especially for PLHIV with multiple comorbidities). To further facilitate the patient visit, PLHIV recommended the check-in function also inform users of approximate waiting times before their appointments and approximate appointment duration. Also, HCPs suggested the addition of a pre-visit checklist with questions regarding reasons for consultation, adherence, and risky sexual behaviors.

3. Responsiveness to patient preferences: Both PLHIV and HCPs enjoyed the possibility of a patient portal’s account settings concerning language (i.e., English or French), security (i.e., password protection, security questions), and data access and sharing. For both participant groups, these account settings seemed easy to adjust. Some participants felt that privacy and confidentiality were ensured, despite their concerns of HIV status disclosure and associated stigma. Participants’ highlighted personal preferences may vary among PLHIV. To further accommodate all users and data sharing preferences, PLHIV recommended the option of synchronizing a patient portal with other online platforms and social media, as well other clinics and specialties.

4. Fulfillment of current or evolving user needs: All participants mentioned that a patient portal and its proposed functions could fulfill several PLHIV needs, especially those that arise from constant changes to healthcare and psychosocial circumstances. Among those mentioned were needs due to changing patient schedules, updating waiting times for check-in, and answering urgent patient questions. Indeed, dynamic two-way communication functions were perceived to grant the app adaptability. As recommendations, PLHIV wished to include features such as a chatbot, a software simulating real conversations, or a discussion forum. Furthermore, both groups wanted PLHIV to be able to provide routine feedback to clinicians through a survey or PROMs, and remote desktop access to a patient portal. All proposed additions were deemed to facilitate communication, highlight PLHIV’s key and emerging concerns, and build the provider–patient relationship. Additionally, HCPs suggested adding a voice function for those who cannot read.

## 4. Discussion

This qualitative study employed focus group discussions to understand the perspectives of PLHIV and their HCPs regarding the benefits and risks of using a patient portal in HIV care. It was conducted to involve stakeholders and account for their needs in the configuration of a stakeholder-approved patient portal. In this Canada–France study, our team conducted ten semi-structured focus group discussions with PLHIV and a diverse group of HCPs and identified five major themes of risks and four major themes of potential benefits. Between each focus group, there was a high level of agreement on the perceived benefits and risks for both PLHIV and HCPs.

This study highlights important perceived risks that should be addressed when seeking to implement a patient portal into HIV care. These include risks related to patient confidentiality, dehumanization of care, and increases in HCPs’ workload. Specifically, several participants were concerned with a patient portal’s ability to maintain confidentiality, particularly when considering HIV stigma. Security and privacy are common concerns with patient portals and are considered a major barrier to their uptake [40,41,42,43,44]. However, past research has found patients to be willing to accept confidentiality risks if they find some benefits like the convenience of online access to their personal health information [45] or to use an effective and useful mobile application [46]. Additionally, as suggested by this study’s findings, measures can be applied to maximize security, including an automated logout after a certain time of inactivity and a two-step authentication process to log in and verify users’ identity.

Our results also indicate PLHIV’s and HCPs both have concerns regarding the nature and quality of care. PLHIV were also concerned that using a patient portal could dehumanize care, fearing that online interactions would replace in-person consultations. In contrast, the prior literature states that patient portals can facilitate more efficient and informed in-person consultations through better-informed discussions and additional channels of communication (for example, text messaging, notifications, etc.) [47]. Nonetheless, HCPs were concerned about an increased workload due to the need to provide additional technical and medical support. These concerns are echoed by HCPs who used patient portals for chronic disease management [14]. However, evidence suggests portals may improve workflow, as they are shown to correct medication errors, promote uptake of preventative services [48], and facilitate communication [17,49]. Workflow may be further enhanced with the incorporation of PROMs [50]. PROMs enable HCPs to optimize care delivery by quickly assessing individual needs and focusing the consultation on priority topics [50,51]. This self-reported data can be efficiently recorded with longitudinal trends to highlight positive and negative symptoms, treatment side-effects, and psychosocial concerns [52]. Engaging stakeholders and providing concrete demonstrations of how a patient portal can be used within HIV care [53] could help address these perceived risks.

This study also noted benefits to be particularly useful for PLHIV who may forget their appointments or to take their medications. These benefits echoed a recently-published systematic review, which examined the effects of patient portals on chronic disease management, including, promotion of medication adherence, enhancement of patient knowledge of their disease, and encouragement of self-management [13]. Several functions were noted to support these benefits, including a calendar, notification, account settings, check-in, PROMs, and education material. Considering the perceived importance of these functions, their integration in a patient portal for HIV care must be considered.

Participants suggested HIV-specific educational material including topics on monitoring their HIV diagnosis, HIV treatment, immigration information, and community-based resources, highlighting stakeholders’ interest in understanding more about their diagnosis and other pertinent social issues. Such educational content was valued by both PLHIV and HCPs to support informed decision-making between PLHIV and their HCPs. Indeed, prior literature has emphasized the capacity of patient portals and its educational material to facilitate shared decision-making with vulnerable patients [41,54,55]. Thus, up-to-date patient portal content and educational material should be considered to optimize the patient portal’s implementation.

Our study has several limitations. Recruitment of PLHIV who identified as Indigenous or women was limited within the CVIS, as there are few Indigenous patients and a greater proportion of men to women at this site. However, biases in PLHIV recruitment were mitigated through maximum variation sampling. Further engagement of underrepresented PLHIV, as planned in the next steps, will be important to ensure the accessibility and usefulness of a patient portal at the study sites. Additionally, this study aimed to recruit at least four individuals per FGD [33]; however, one group consisted of only three HCPs as two participants were unable to attend. The FGD with three HCPs was limited in size, thus, limiting the group’s perspectives and experiences; however, this study aimed to mitigate this through a relatively large number of ten focus groups to achieve thematic saturation [39].

## 5. Conclusions

This study employed a qualitative methodology to understand the perspectives of PLHIV and HCPs from Canada and France regarding the benefits and risks of using a patient portal for HIV care. The perspectives of PLHIV and HCP provided valuable insight for optimizing the benefits of a patient portal and addressing its perceived challenges when considering its implementation.

## Figures and Tables

**Table 1 jpm-12-00314-t001:** Functions of a patient portal that were discussed with all study participants.

Function	Description
Account settings	Account settings allow patients to customize the portal to their preferences, including language and font size.
Calendar	A calendar function allows patients to view their upcoming appointments.
Check-in	Patients can notify HCPs of their arrival at the clinic and receive approximate waiting times.
Consultation notes	HCP’s (clinicians and nurses) consultation notes are displayed for the patient to view.
Diagnostic test results	Patients may access their laboratory test results, including a longitudinal display of trends.
Education material	Personalized education material with explanatory content is provided in text and/or video formats.
Navigation tool	An intra-hospital map orients patients to their clinic or room.
Notifications	Notifications advise patients when it is time for their appointment and if patients are next in line for treatment or their appointment.
PROMs	Self-reported electronic questionnaires are administered to patients to identify pertinent symptoms, concerns, or needs.
Texting	Direct text messaging allows HCPs to send announcements and messages to their patients.
Treatment Planning	Patients can view their personal treatment plan and planning status information.

**Table 2 jpm-12-00314-t002:** Characteristics of participating people living with HIV and healthcare providers.

	People Living with HIV (*n* = 28) %	Healthcare Providers (*n* = 31) %
Focus group		
MUHC (English)	7 (25)	
MUHC (French)	5 (18)	
MUHC (French)	8 (29)	
Saint-Antoine Hospital (French)	8 (29)	
MUHC (English)		6 (19)
MUHC (French)		6 (19)
MUHC (French)		4 (13)
Saint-Antoine Hospital (French)		7 (23)
Non-MUHC Montreal (French)		8 * (26)
Age group		
20–39	6 (21)	8 (26)
40–59	15 (54)	14 (45)
60–79	5 (18)	3 (10)
Missing	2 (7)	6 (19)
Gender		
Male	18 (64)	10 (32)
Female	9 (32)	20 (64)
Other	1 (4)	1 (3)
Sexual Orientation		
Heterosexual	15 (54)	N/A
Men who have sex with men	10 (36)	
Bisexual	2 (7)	
Other	1 (4)	
Preferred language		
French	17 (61)	18 (58)
English	6 (21)	7 (23)
French and English	4 (14)	6 (19)
Other	1 (4)	0 (0)
Occupation		
Physician	N/A	13 (42)
Pharmacist		8 (26)
Nurse		6 (19)
Social Worker		2 (6)
Clinical care coordinator		2 (6)
Ethnicity		
African	13 (46)	N/A
Caucasian	11 (39)	
Latino	4 (14)	

* Non-MUHC focus group discussions with healthcare providers were carried out in two groups, consisting of five and three individuals, respectively. Percentages are rounded to the nearest whole number and thus, totals may not equal 100%. Abbreviations: N/A = not applicable; MUHC = McGill University Health Centre.

**Table 3 jpm-12-00314-t003:** Examples of participant quotations on the potential benefits and risks of a patient portal in HIV care.

Potential of Patient Portal Use	Participant Quotations	
	People Living with HIV	Healthcare Providers
Risks		
Breach of confidentiality	You should talk with your doctor, because maybe you don’t understand if you share this note (to other HCPs). So, I think it’s extremely sensitive and it should be controlled, or you should be conscious about what you are sharing. (European Man, 28 years old, heterosexual) Log in on your (patient portal) and you will be able to see whatever. For example, imagine if you lose your phone and someone has access. (African male, 44 years old, heterosexual) From (your) lab results everyone can know that you are under HIV treatment … that your viral load is that, your CD4 is that. (White male, 61 years old, heterosexual)	You can’t have an application that stays open, let’s say someone has their phone and someone else comes across and just looks at the phone and see that it’s open. (Pharmacist) There has to be some programming where if you haven’t used it in 30 s you are logged out. (Pharmacist) There are much more concerns surrounding privacy and confidentiality and the high rate of psychiatric illness in our population and drug use…There are issues about having that (consultation notes) accessible to others…(Physician)
Stress or uncertainty	(If) I cannot interpret the results in the right way… I start stressing for the next two weeks before I see my doctor. (African male, 40 years old, Heterosexual) If there is a snowstorm and the internet is interrupted, I don’t know how fast the changes of the dates or appointments would be. (African woman, 30 years old, heterosexual)	I would like to have… a portal where the person has reliable (health) information. (as misinformation can cause stress) (Physician) Somebody might have a viral load of 24 and it can cause huge anxiety. (Nurse)
Contribution to the digital divide	Yes, but they will still need to have a somewhat sophisticated phone if they want to have the application. (White man, 70 years old, Men who have sex with men) I don’t want to have it as an app on my phone, but I want to have it at home on my computer. (Since not all PLWH have smartphones) (African man, 44 years old, heterosexual) There are some phones that are not compatible with it (a patient portal’s features) (African man, 40 years old, heterosexual)	How many refugees have a phone? (Nurse) (A patient portal) is based on the fact that you have to have a phone. (Social worker) It will only be those who have access to a smartphone that can use it. (Nurse)
Dehumanization of care	I would have liked the doctor to reassure me, to meet me at least to tell me (about their laboratory results) (African woman, 58 years old, heterosexual) You cannot call somebody to counsel you on the phone. You need to see the person physically and emotionally so I’m wondering how it can be put into (a patient portal). (African man, 51 years old, heterosexual)	I don’t think there is any other replacement other than a person (Physician) It’s warmer when someone who knows you greets you, than a machine. (Physician)
Increase in healthcare provider workload	If you are talking to her (the clinician) about something and then it becomes long, it will be like 15 min with you for her she’s not getting paid or he is not getting paid (if using a patient portal to communicate). (African woman, 30 years old, heterosexual)	In addition to having an application where I would have to respond to concerns in real time, I definitely don’t want that, because it will make our lives unmanageable. (Physician) It will need a lot of support because we are not accessible 24 h a day. (Physician)
Benefits		
Improvement in HIV self-management	The (calendar) is really good… because we know that one or two weeks before you see the doctor you need blood tests. (African man, 40 years old, heterosexual) Yes (on sharing consultation notes) if you are not dealing with the people here… If it (the clinic visit) happens in Toronto, you need to go to the hospital and the doctor will be asking a lot of questions. (White male, 61 years old, heterosexual) I would need in a patient portal probably just general information like what’s new with research, what new discoveries have been made recently. (Latino man, 46 years old, bisexual)	If this (notification function) can be integrated into the patient’s application, it will greatly simplify medication renewal. (Pharmacist) I think it’s (a patient portal) useful. People are unfortunately not always informed, or at least not always aware of their problems, or they minimize them (for access to consultation notes and PROMs). (Physician) I think it’s great for patients to know and they can show it (consultation notes) to whoever in their family. (Social worker) It would be good if the patients had a list of all the community organizations, their roles, and their contact information, and where to go if you are a woman with HIV. (Pharmacist)
Facilitation of patient visits	I have so many things to tell her at one time. Of course, she takes her time as a doctor but then to me I feel like the time is not enough. That portal if it can have it [lab results] in advance. (African woman, 50 years old, heterosexual) That (appointment management) would save calling the hospital to make appointments. (African Woman, 54 years old, heterosexual) Yes, it’s (check-in) really better than sitting in queue and not knowing when your name is going to be called. (African woman, 30 years, heterosexual) I guess it’s a good application, a good function (navigation tool) because we are not familiar with every section of the hospital. (White man, 61 years old, heterosexual)	It’s (calendar and notifications) still useful, because there are a number of patients who miss their appointments… (Physician) Speed of registration (using a check-in function), perhaps needing a little less staff for this kind of work, avoiding queues. (Physician) Why is he taking this (medication)? Anyway, it (consultation notes) would help us a lot, that’s for sure, as professionals. (Pharmacist) Everything (should) be in the app in terms of where your appointment is ((navigation tool) … what time, etc. (Clinical coordinator)
Responsiveness to patient preferences	Anything additional other than English and French would be good. Especially for elderly people that come in the country and they don’t even have that time to start learning the languages. (African woman, 30 years old, heterosexual) To be able to configure only the things you need because you can have everything there. Maybe it’s overwhelming for people that only want to use a small part. (European man, 28 years old, heterosexual) (I) suggested facial recognition technology but there is a possibility that there are some phones that are not compatible with it. (African man, 40 years old, heterosexual)	It (automated log out) should be automatic because… patients don’t remember to log out. (Nurse) It (account settings) just should be able to enable or disable certain aspects if they want or don’t want it. (Physician) (We should be) able to choose which notifications you are getting so you don’t get bombarded. (Pharmacist) Don’t be afraid to use pictograms, like pictures, to describe what to do. (Social worker)
Fulfillment of current or evolving user needs	Not only to have it on your phone but also on your computer so that you can have it send you a text or message to say “Ok, you have medication” or “Your next appointment will be on that date”. (African woman, 31 years old, bisexual) Sort of like chatting (on a chat box) like we do with friends. I am missing that on the portal. (African woman, 50 years old, heterosexual) Yes, ongoing feedback (patient feedback survey) is how we can improve it (a patient portal). (African man, 51 years old, heterosexual) It (appointment management) would engage people in other complementary ways. (White woman, 47 years old, heterosexual)	My personal opinion is that patients would like (PROMs) too because it shows we care about those issues, and they feel actually uncomfortable bringing them up. (Physician) We could have some kind of evaluation of the overall satisfaction with the clinic care and where are the areas where they would like improvement. (Physician) I think it would be useful for the carer to have, for example, a collection of information on the weeks or months that preceded, of data perhaps, of difficulties that the patient encountered. (Nurse) From diabetes to cancer to HIV… (Synchronizing data to other clinics) It could be just a way of improving the medical system and updating it to new technologies. (Pharmacist)

## Data Availability

The data presented within this study are not publicly available due to privacy and ethical considerations.

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
