# Peer review of "Understanding the Risks and Benefits of a Patient Portal Configured for HIV Care: Patient and Healthcare Professional Perspectives"

_jpm, 2022, doi:10.3390/jpm12020314_

Round 1
Reviewer 1 Report
Authors reported a research article with the aim of elucidating he the perceived risks and benefits among peaole who live with HIV and healthcare professionals of patient portal use in HIV clinical care. The strength of the study is novelity in the topic, the weakness is a smal sample size. However, authors presented intriguing results that clarified that both risks and benefits of HIV care can be identified by a patient portal. I would like t put forward several items to discuss.
- The study was conducted in 2 cites, so, it would be great to know whether the results were similar in both centres.
- Data coding was not thoroughly reported. Please, extend this subsection and give clear explanation of how the data were collected.
- Authors did not report in which way data were analyzed. Please, go through with the results and support them with the descriptive statistics.
Author Response
Please see the attachment, thank you.

Reviewer 2 Report
This qualitative study used focus group discussions to understand the perspectives of people living with HIV and their healthcare professionals on the benefits and risks of using a patient portal in HIV care. Five main themes of risks and four main themes of potential benefits were identified in this study. This study highlights important perceived risks that should be addressed when trying to implement a patient portal in HIV care. The study is of interest to the readers of this magazine. Could be improved, using a simpler form of displaying the data and results obtained, as well as some suggestions:
The introduction must be improved by adding other similar articles present in the literature and the results obtained by them.
In Context and participants: (clinicaltrials.gov identifier: NCT02586584), must be added as a reference.
Complete table 2, also inserting the percentage values ​​in brackets.
Author Response
Please see the attachment, thank you.

Round 2
Reviewer 1 Report
Authors submitted a revised version of the article along with comprehensive explanations according to the reviewers' recommendations and correctioms made. I have no serious flaws to the article in its revised version.
Reviewer 2 Report
the manuscript has been improved